

# Hydrometeorological conditions preceding wildfire, and the subsequent burning of a fen watershed in Fort McMurray, Alberta, Canada

15   Matthew C. Elmes*[1], Dan K. Thompson[2], James H. Sherwood[1], Jon S. Price[1]

[1]Dept. of Geography and Environmental Management, University of Waterloo,
Waterloo, Ontario, Canada, N2L 3G1
[2] Natural Resources Canada, Canadian Forest Service, Northern Forestry Centre, 5320 122 Street
Northwest Edmonton, Alberta, Canada, T6H 3S5

*Correspondence: elmes.matt@gmail.com





**Abstract.** The destructive nature of the ~590,000 ha Horse River Wildfire in the Western Boreal Plain (WBP), northern Alberta in May of 2016 motivated the investigation of the hydrometeorological conditions that preceded the fire. Historical climate and field hydrometeorological data from a moderate-rich fen watershed were used to identify a) whether the spring 2016 conditions were outside the range of natural variability for WBP climate cycles; b) explain the observed patterns in burn severity across the watershed; and c) identify whether fall and winter moisture signals observed in peatlands and lowland forests in the region are indicative of fire susceptibility. Field hydrometeorological data from the fen watershed confirmed the presence of cumulative moisture deficits prior to the fire. Hydrogeological investigations highlighted the susceptibility of fen and upland areas to water table and soil moisture decline over rain-free periods (including winter), due to the watershed's reliance on supply from localized flow systems originating in topographic highs. Subtle changes in topographic position led to large changes in groundwater connectivity, leading to greater organic soil consumption in wetland margins and at high elevations. The 2016 spring moisture conditions measured prior to the ignition of the fen watershed were not illustrated well by the Drought Code (DC) when standard overwintering procedures were applied. However, close agreement was found when default assumptions were replaced with measured duff soil moisture recharge and incorporated into the overwintering DC procedure. We conclude that accumulated moisture deficits dating back to the summer of 2015 led to the dry conditions that preceded the fire. The infrequent coinciding of several hydrometeorological conditions, including low autumn soil moisture, a modest snowpack, lack of spring precipitation, and high spring air temperatures and winds, ultimately led to the Horse River wildfire spreading widely and causing observed burn patterns. Monitoring soil moisture at different land classes and watersheds would aid management strategies in the production of more accurate overwintered DC calculations, providing fire management agencies early warning signals ahead of severe spring wildfire seasons.

## 1 Introduction

The sub-humid Athabasca oil sands region (AOSR) of the Western Boreal Plain (WBP) comprises a mosaic of small lakes, forested uplands, and wetlands primarily as peatlands (Devito et al., 2012). Bogs are defined as ombrogenous peatlands, receiving water exclusively from atmospheric sources (Ingram, 1983). Conversely, fens are geogenous, receiving water from both atmospheric and surface water and/or groundwater sources. In the WBP, fens are distinguished into three primary types (poor, moderate-rich, and extreme-rich) based on differences in water chemistry and indicator plant species (Vitt et al., 1995). In the AOSR, moderate-rich fens are the primary peat-forming wetland (Chee and Vitt, 1989). The hydrology of bog, poor fen (Ferone and Devito, 2004; Wells et al., 2017) and saline fen (Wells et al., 2015a, 2015b) systems have been studied in the WBP, however, the hydrology of moderate-rich fen systems in the AOSR remains largely unexplored.



Water availability in the WBP is constrained by annual precipitation rates that are typically less than potential evapotranspiration (PET) demands (Marshall et al., 1999; Bothe and Abraham, 1993). Consequently, the timing, frequency, and magnitude of wildfires is dictated by variability of the hydrometeorological conditions over the growing season (Abatzoglou and Kolden, 2011; Flannigan and

Harrington, 1988), where moisture deficits accumulate in upland duff (Keith et al., 2010) and near-surface peat horizons (Lukenbach et al., 2015) over extended dry periods. Wildfire liberates large quantities of terrestrial carbon stock held within WBP peatlands, estimated at 4700 Gg C released per year, from Continental Western Canada alone (Turetsky et al., 2002); the peat is vulnerable to combustion and deep smouldering (Benscoter et al., 2011; Turetsky et al. 2011). Over the past decade,

there has been increasing concern over longer fire seasons in Alberta (Wotton and Flannigan, 1993; Flannigan et al., 2013; Kirchmeier-Young et al., 2017), and an increase in large high-intensity wildfires (Tymstra et al., 2007), and total burned area each year (Turetsky et al., 2002).

The majority of summer wildfires are ignited by lightning (Tymstra et al., 2005), when wildfire behaviour can be predicted by drying signals in shallow forest duff horizons with relatively simple

drying mechanisms (Wotton et al., 2005). Unlike summer fires, spring wildfires usually have human caused ignition sources (e.g. recreational vehicle exhausts or unextinguished cigarettes), and are harder to predict given that widespread fires occur regardless of the presence of moisture deficits (Amiro et al., 2009). These spring fires therefore possess less obvious antecedent moisture signals, given that they occur post-snowmelt, an important rewetting period for wetlands and forested uplands in the region

(Smerdon et al., 2008; Redding and Devito, 2011).



In Canada, early spring fire susceptibility is typically predicted with the Canadian Forest Fire

Weather Index (FWI) System, a component of the Canadian Forest Fire Danger Rating System

(CFFDRS) (Lawson and Armitage, 2008). The Drought Code (DC) is a component of the FWI which

applies to slow-drying deep forest organic layers often found throughout the WBP, layers that can

enhance wildfire intensity (Van Wagner, 1987). The DC is a semi-physical model which uses

precipitation inputs and predicts water loss as a function of daily noon temperature and day length, to

estimate the moisture content of deep organic layers that typically dry logarithmically based on an

estimated 53-day period required to lose two thirds of held moisture (Lawson and Armitage, 2008).

Values of the DC range from 0 (saturated soils at surface) to over 800 (residual soil moisture only),

representing the origins of the index as representing the slow drying of stored water in Pacific coastal

slash fuels (Turner, 1972). DC values have been related to peatland water table (Waddington et al.,

2012) as well as the extent of peatland burned area (Turetsky et al., 2004). These DC calculations,

although based on typical wetting and drying rates of relatively deep upland fuels (Lawson and

Armitage, 2008) and regarded as general estimates, can be important predictors for fire managers

immediately following snowmelt, especially when additional soil moisture information is lacking.

Given the large moisture deficits that can develop in deeper upland organic layers, the DC is

overwintered to incorporate the effect of fall DC and winter precipitation into the next year's starting

value. Overwintering calculations generally include estimates of total winter precipitation from nearby

climate stations, along with two estimated coefficients, which include a carry-over effect to adjust for

antecedent (fall) moisture conditions, and the wetting efficiency fraction of the snowpack to the specific



soil layer (Lawson and Armitage, 2008). These coefficients, however, can be ignored if direct measurements of recharge into forest soils are available.

During the spring of 2016, the ~590,000 ha Horse River Wildfire spread into the city of Fort McMurray, and subsequently advanced across the boreal mosaic of mixedwood uplands and peatlands.

The destructive nature of the Horse River Wildfire motivated the investigation of the hydrological and meteorological conditions that led up to the fire. Currently, it is unknown if the exceptionally warm and dry conditions were also manifested by significant moisture deficits in the peatland watersheds surrounding the community.

This study capitalizes on an opportunity to explore pre-fire hydrometeorological data obtained

from Poplar Fen from 2011-2016, an instrumented moderate-rich fen watershed that burned on May 17, 2016. The specific objectives of this research are 1) to use a combination of historical climate and field hydrological data to characterize the hydrometeorological conditions preceding the burning of a moderate-rich fen watershed to determine whether these conditions were outside the range of natural WBP climate cycles; 2) to use these hydrological data to explain the observed patterns in burn severity

across the watershed; 3) to identify whether hydrological data and hydrogeological setting parameters of the watershed can serve as indicators of deep smouldering and combustion risk. This study provides a useful means of explaining why watersheds in the region are especially vulnerable to wildfire in the early spring, and how management agencies can better detect these early moisture signals that are a potential indicator of future high-intensity spring wildfires.



## 2 Study Site

Situated within the Athabasca region of the Boreal Plains ecozone (Ecoregions Working Group, 1989), 'Poplar Fen' (56°56'N; 111°32'W Fig. 1) is a ~2.5 km$^2$ treed moderate-rich fen watershed, located 25 km north of Fort McMurray, Alberta (Fig. 1). This watershed is characterized by low relief topography

(~10 m), with undulating sand/gravel uplands, and interconnected moderate-rich fens. Average annual air temperature (1981-2010) is 1°C; average annual precipitation is 419 mm, with ~75% falling as rain (Environment Canada, 2017).

The fen areas at Poplar Fen are underlain by heterogeneous deposits of fine-grained silts of relatively low hydraulic conductivity that constrict recharge to an underlying sand/gravel layer and

10 favour saturated peat-forming conditions. Maximum peat depth ranges from 1.2–3.0 m, decreasing to <0.5 m along the margin tracts between fen and upland. Ground surface and water table elevations generally decrease from upland to margin to fen, and from southeast to northwest.

Tamarack (*Larix lariciana*) and black spruce (*Picea mariana*) are the dominant tree species within moderate-rich fen areas, with a surface cover of the mosses *Tomenthypnum nitens, Aulacomnium*

*palustre, Pleurozium schreberi,* and from the genus *Sphagnum* (*S. fuscum* and *S. capillifolium*). Margins are characterized by a sparse *P. mariana* overstory, with *S. fuscum* and the feathermosses *Hylacomium splendins* and *P. schreberi*. Upland areas are dominated by *P. mariana* and feathermosses in riparian zones, with jack pine (*Pinus banksiana*) and aspen (*Populus tremuloides*) mixedwood overstory and lichen ground cover, in topographically higher areas.



## 3 Methodology

### 3.1 Historical data collection

A 20-year record of meteorological data were obtained from Alberta Agriculture and Forestry through the Alberta Climate and Information Service (Alberta Agriculture and Forestry, 2017). This included

daily values of precipitation (rainfall and snowfall) and air temperature, which were estimated for the Poplar Fen area (township: T092R10W4) using an inverse-distance weighting interpolation procedure (IDW). Data from 2-7 stations were used for the IDW over the 20-year period with the nearest station (Mildred Lake; Fig. 1) located ~12 km from Poplar Fen. Rainfall and snowfall were totaled for every hydrological year (Oct. 01–Sep. 30). Additional average daily wind speed and relative humidity values

were obtained from the Mildred Lake climate station for the 2015-16 winter and early spring (Alberta Agriculture and Forestry, 2017). A seven-year record of snow-water equivalent (SWE), the depth of water contained within the snowpack, was also obtained from a snow pillow located at Gordon Lake, ~70 km from Poplar Fen (Alberta Environment and Parks, personal communication). This record provided information on SWE accumulation/ablation as well as peak SWE prior to snowmelt from Oct.

2009 to Apr. 2016. Snow-free conditions were estimated for each year by identifying the day when <20% of the snow mass was remaining.

### 3.2 Field data collection

Hydrological data were collected between 2011 and 2016. Initial instrumentation included a water table monitoring well in a fen area (NW fen) located in the northwest section of the watershed, and a well in

the adjacent margin area (NW margin), located ~140 m south of the fen well (Fig. 1) at an elevation ~0.65 m greater than the fen. NW fen water table was monitored from June 2011 to May 2016 using a



logging pressure transducer (Schlumberger D501 Mini-Diver). In spring 2015, additional groundwater

monitoring focused on two fen areas located at contrasting low (lower) and high (upper) topographic

elevations, which varied by ~0.7 m (Fig. 1). Groundwater monitoring nests were installed in the lower

(*n*=4) and upper (*n*=3) fen areas and adjacent margins (*n*=4) (Fig. 1). Screened wells and piezometers

(20 cm screened intake) were constructed from PVC (2.5 cm I.D.) pipe and installed into the different

substrates in grouped nests. Nests typically comprised a fully-slotted well, with piezometers installed in

mid-peat (0.6–0.75 m depth) and underlying mineral (1.25–1.85 m depth). Logging pressure transducers

were installed in a well and in a piezometer situated in the underlying mineral layer, for one nest each in

the lower and upper areas (Fig. 1). Nests were measured manually on a weekly basis between May–

August, 2015, and again in early October. Vertical hydraulic gradients were calculated between water

table and hydraulic head in the underlying mineral layer for each nest in the lower and upper fen areas,

using standard methods (Freeze and Cherry, 1978). Fen ground temperatures were monitored within

close proximity to the NW fen well between fall 2012 and summer 2016 using two thermocouple

arrays, buried at 0.1 and 0.2 m depth below surface. Temperatures were logged half hourly and daily

averages were calculated for each depth.

Precipitation was measured in an open area of the site with a logging Onset RG3-M tipping

bucket rain gauge; missing daily totals (Oct–May) were supplemented with interpolated rainfall data for

the Poplar Fen area (Alberta Agriculture and Forestry, 2017). Between Mar. 21 and Apr. 19, 2016,

snow surveys were conducted using a Meteorological Service of Canada (MSC) snow tube.

Measurements of snow depth were taken at 178 locations, ~ 10 m apart along a zig-zag transect that

extended through all major land class for Poplar Fen (Fig. 1). SWE was recorded every ~20 m. Peak





SWE was represented by the first snow survey on Mar. 21, 2016 and an area-weighted SWE

contribution for each land class was estimated from the proportional area for each respective class.

Volumetric water content (VWC) was recorded half-hourly from June 2015 to May 2016 in

upland duff and margin peat soils, with arrays of Stevens Hydra-II probes (Fig. 1;2). Two weeks of data

(May 2–May 17, 2016) could not be salvaged due to fire damage to the logger. The probes were

calibrated in-laboratory to the respective soil types.

**3.3 Drought Code**

The Drought Code (DC) was calculated using the 'cffdrs' package in R statistical program (R Core

Team, 2016) for the 2015 growing season using data obtained from the Mildred Lake climate station

(Alberta Agriculture and Forestry, 2017). This included noon measurements of air temperature and

cumulative precipitation from the previous 24 hours. The DC was started on Apr. 12, 2015, following

three days of noon temperatures of 10°C or higher, using default presets, including a starting DC of 15.

The starting DC becomes less imperative over a fire season as it will be corrected after sufficient

rainfall (Alexander, 1982), thus, an overwintering procedure is essential for improving accuracy

predominantly in the early months of a fire season. The DC was run until Oct. 31, 2015 (a standardized

end date) and the code then overwintered for spring 2016 using a range of different approaches

following methods outlined by Van Wagner (1987). The startup moisture equivalent ($Q_S$) of the DC was

determined by Eq. (1):

$$Q_S = aQ_f + b(3.94r_w), \tag{1}$$



where $Q_f$ is the moisture equivalent of the DC on October 31, 2015, $r_w$ is total winter precipitation (mm), and *'a'* and *'b'* are coefficients chosen to estimate the carry-over fraction of last fall's moisture, and estimate the fraction of snowmelt retained in the duff layer, respectively. $Q_f$ is calculated by Eq. (2):

$$Q_f = 800 \exp(-DC_f/400), \quad (2)$$

where $DC_f$ is the final DC value on Oct. 31, 2015. The startup DC value can then be calculated from Eq. (3):

$$DC_S = 400 \ln(800/Q_S), \quad (3)$$

The values for *a* and *b* in equation 1 are typically determined by provincial fire management
agencies (Lawson and Armitage, 2008). Though organic soil moisture data are available in this instance, in this study we examine both the observed soil moisture data, in addition to variations on DC start and overwinter values using less detailed information, to compare predictions of organic soil moisture at the time of the fire made without the benefit of in-situ observations.

Startup and overwinter upland duff DC were calculated four different ways (Table 1), each
reflecting specific information of the hydrometeorological environment. For scenario 1, startup DC was estimated for the upland duff from a linear regression between DC and measured duff VWC from Jun. 27–Oct. 31, 2015, and calculated based on the starting VWC for Apr. 19. Scenarios 2-4 were then calculated with the overwintering procedure (Equations 1, 2 and 3). For scenario 2, the startup DC was calculated using total winter precipitation values obtained from the Fort McMurray airport climate



station and default carry-over and wetting-efficiency values (0.75) from the cffdrs package (Lawson and

Armitage, 2008). For scenario 3, the startup DC was calculated from peak SWE data from snow survey

data of Poplar Fen and carry-over (0.5) and wetting-efficiency (1.0) values used by Alberta Agriculture

and Forestry. Scenario 4 applied the directly measured duff recharge (a mm value input, inferred from

5    the upland duff site moisture probe) to the overwintering procedure, which eliminated the need for a

precipitation value as well as estimates of carry-over and wetting-efficiency. Following these methods,

four differing startup DC values were generated for the upland duff. The DC was then calculated four

times, corresponding to each startup DC value, starting at Apr. 19 and were ran until May 17, 2016.

### 3.4 Burn depth and fuel consumption

10    Measurements of burn consumption of organic soils were made in fen, margin, and upland areas that

burned using differential GPS (Leica GS14 GNSS) survey data from well stick-up (length of PVC

above ground surface) elevations; the difference between soil surface elevations at piezometer nests

were compared between 2015 and post-fire 2016. This included nests from Poplar Fen additional (5 fen,

5 margin, and 10 upland nests) to those identified in Fig. 1 (not shown). Average vertical elevation

15    (surface) change was calculated for each nest location. Depth changes were averaged for burned fen,

margin, and upland organic soils, and these depths multiplied by previously measured average bulk

density values for each soil type to estimate terrestrial fuel loss.



## 4 Results

### 4.1 Hydrometeorology

Precipitation observations from 1996-2016 interpolated for Poplar Fen averaged 380±17(SE) mm total

5   precipitation with 284±15 mm falling as rain and 96±6 mm as snow (Table 2). For the four hydrologic

years of high burned area in the spring, total winter snowfall was below average for all years except for

1997-98. The lowest total snowfall was measured in hydrologic years 1998-99, 1999-00, and 2008-09,

all years with low burned area in the spring. Peak SWE from Gordon Lake snow pillow from 2009-

2016 (Fig. 3) averaged 120±10 mm. Peak SWE prior to snowmelt in hydrological years 2010-11 and

2015-16 was not especially low, and despite the modest SWE available for melt, the snow-free day of

year (80% of peak SWE melted) during these years was not significantly earlier than the other 5 years

on record. Total rainfall was below average in only two (1997-98 and 2010-11) of the four hydrological

years with large spring burned areas, the bulk of precipitation occurred in the summer for all four years

(Table 2). Cumulative post-melt rainfall until May 15 averaged 25.5 ± 3.3(SE) mm between 1997 and

2016 (Fig. 3). Three of four hydrological years with high burned area in the spring were below the 20-

year average rainfall, 2001-02 being the lowest and 1997-98 just above average. In 2015-16, only 8.5

mm of rain fell following snowmelt prior to ignition of the Horse River Wildfire, and only 0.3 mm fell

over the next two weeks leading up to the burning of Poplar Fen watershed (Fig. 3). The hydrological

year with the lowest early spring cumulative rainfall in the 20-year record was 2007-08 (1 mm), a year

of low burned area in the spring (Natural Resources Canada, 2017). However, during this year a total of



151 mm of snow fell in the area, 55 mm more than the 20-year average (Table 2), and snow-free

conditions were not reached until Apr. 30.

Over the 2015-16 winter (mid Oct. – mid Apr.), average air temperatures were -6.5°C, 2.9°C

warmer than for the 20-hydrologic year (1996-2016) average (-9.3°C). Periodic warm spells were

observed in late January and February, when air temperatures rose above freezing for several

consecutive days (Fig. 4a). Manual snow measurements yielded an area-weighted average peak SWE of

105 mm (Fig. 4d) on Mar. 21, 2016, 2 mm higher than the peak measured at the Gordon Lake snow

pillow. In spring 2016, the primary snowmelt period occurred between Mar. 21 and Apr. 19. Air

temperatures did not deviate far from the 20-year normal during this time, with the exception of Mar.

27-30, when daily maximum air temperatures in the area rose to over 9°C. The strongest deviation prior

to the fire was measured following snowmelt when maximum daily air temperatures reached 27°C and

33°C in April and May, respectively. At this time, average daily relative humidity decreased (Fig. 4b)

and average daily wind speeds exceeded 20 km hr$^{-1}$ prior to the fire's ignition (Fig. 4c).

### 4.2 Hydrology

The NW fen (Fig. 1) water table range was ~0.79 m, (+0.12 m to -0.66 m) between Jun. 08, 2011 and

May 17, 2016 (Fig. 5). Average NW margin water table was 0.32 m lower than NW fen between 2011

and 2015. Both NW fen and margin exhibited relatively low water tables (dry conditions) at the

beginning, increased water table in the middle years (wetter conditions), and lower water tables in a

drying period towards the end of the 5-year record. The late fall and early spring NW fen water table

was near or above ground surface in hydrological years 2012-13 and 2013-14 (Fig. 5), which

corresponded with delayed ground thawing at 0.1 and 0.2 m peat depths until mid-May. Conversely,



year 2014-15 water table was ~0.2 m b.g.s. in the fall and at the surface in the early spring, which began

to decline rapidly in June (Fig. 5). This hydrological year corresponded with delayed ground thawing

until mid-May at only 0.2 m peat depth. Furthermore, between 2011-2015, NW fen water table

underwent periods of decline over the summer in all years except 2013. By early fall, the NW fen water

table in all five years reached an annual low, and in 2012-2014, rose in the late fall prior to freeze-up.

Conversely, rainfall was not sufficient in 2011 and 2015 to raise the fall NW fen water table above the

low levels observed in the summer (Fig. 5).

The 2015-16 hydrologic year began with water levels that were among the lowest in the six-year

record (Fig. 5). By the end of winter, all manually-surveyed fen monitoring wells were empty of water

(water tables >0.8 m b.g.s.). The comparison of fall 2015 logged water levels to manual winter 2016

observations (before snowmelt recharge and before pressure transducers were installed for the 2016

field season) evidenced an additional 0.12-0.26 m water table decrease, demonstrating mid-winter water

table decline and drying of overlying peat substrate. Ground thawing at 0.1-0.2 m depth occurred in

mid-April (earlier than 2013-15) toward the end of snowmelt, and at this time (Apr. 16, 2016) the NW

fen water table had increased to 0.67 m b.g.s. The remaining snowmelt period initiated a water table rise

of 0.46 m to 0.21 m b.g.s. on May 03, which then decreased in the total absence of rainfall to 31 cm

b.g.s on May 17, the day that Poplar Fen area burned over (Fig. 5).

To examine how fen areas of varying topographic position were wetting and drying over the

2015 growing season, water table and hydraulic gradients were compared between the contrasting upper

20   and lower fen areas (Fig. 6). Average water table depth below surface differed by 0.05 m between upper



(0.22 ± 0.05(SD) m) and lower (0.17 ± 0.04(SD) m) fen nests. In both areas, hydraulic head in

underlying mineral layers mirrored the water table profile (Fig. 6). Vertical hydraulic gradients (a

metric of groundwater recharge/discharge) in both upper and lower fen areas were strongest when water

tables were highest, and weakened (less groundwater recharge to the fen) during periods when rainfall

was less abundant. Throughout the entire monitored period, the lower fen nests had the strongest

average hydraulic gradients (0.021 ±0.008(SE)), showing groundwater discharge that remained positive

throughout the measurement period. Conversely, upper fen nests had weaker hydraulic gradients (-0.007

±0.004(SE)), which experienced flow reversals (downward), and were negative throughout most of the

year. Margin areas exhibited the lowest water tables, as well as hydraulic gradients (-0.03 ±0.03(SE))

(recharge), over the 2015 growing season (Fig. 6).

Between June and October 2015, duff and margin peat VWC (both at ~0.2 m b.g.s.) averaged

0.33 and 0.41 $m^3$ $m^{-3}$, respectively, with a higher coefficient of variation for duff (0.21) compared to

margin (0.06) peat (Fig. 7). The duff experienced extended drying periods in the summer-fall, and by

freeze-up, reached the minimum VWC for 2015 (0.24 $m^3$ $m^{-3}$). Margin peat VWC had also reached a

minimum by fall (0.39 $m^3$ $m^{-3}$), however, values were similar to late spring 2015 VWC (~0.42 $m^3$ $m^{-3}$).

During winter 2015-16 (Oct. 31 - Mar. 21), VWC in the duff and margin peat decreased an additional

0.06 and 0.03 $m^3$ $m^{-3}$, respectively, and following snowmelt, increased from 0.19 to 0.32 $m^3$ $m^{-3}$ and

0.36 to 0.38 $m^3$ $m^{-3}$, respectively. Two weeks prior to the Horse River Wildfire, upland duff and margin

peat VWC were 0.31 and 0.37 $m^3$ $m^{-3}$, respectively (Fig. 7), and likely continued to decrease prior to the

fire in the absence of rainfall.



### 4.3 Drought Code

Following the first month of start up in 2015, the DC illustrated an inverse relationship with upland duff VWC ($R^2 = 0.88$) (Fig. 7); the dry conditions caused DC to increase from 18 to 496, between Apr. 12– Oct. 31. VWC on Oct. 31 and the corresponding DC were chosen to represent the final fall moisture and

DC equivalent values for the various overwintering DC calculations. The overwintering period ran from Oct. 31, 2015 to Apr. 19, 2016, the day following three consecutive days with noon air temperatures ≥12°C. For scenario 1, the 2016 spring start up DC was predicted based on the relationship between upland duff DC and VWC in 2015, and a 2016 spring start up DC of 357 was estimated given a starting soil moisture of 0.37 $m^3$ $m^{-3}$ (Fig. 7). Scenarios 2 and 3 produced startup values using the overwintering

procedure with standard carry-over and wetting-efficiency coefficients, resulting in start up DCs of 242 and 212 respectively. Scenario 4 used the overwintering procedure with no precipitation values or default coefficients, rather with directly measured duff recharge from Oct.31 to Apr. 19. Snowmelt increased duff VWC by 0.13 $m^3$ $m^{-3}$ in the ~0.25 m thick soil horizon, resulting in 32 mm of recharge (35% of estimated upland snowfall), yielding a startup DC of 321. DC was then calculated from Apr. 19

to May 17, 2016, and all starting DCs increased 131 units over that time period (Table 3).

### 4.4 Burn depth and fuel consumption

Greatest depth of burn was measured in the margins (0.13±0.01 m) with lower (0.10±0.02 m) burn depths measured at upland locations (Table 4). Burn depth values in burned fen areas were 78-83% lower than margin and upland areas. Estimated fuel consumption rates (depth of burn x average bulk

density) generally echoed the trends in surface change with slight differences due to higher bulk density measured in margin peat. No surface changes or fuel consumption were observed in the lower fen area.



## 5 Discussion

### 5.1 Pre-fire meteorology

Within the Boreal Plain region of northeastern Alberta, average precipitation is less than potential evapotranspiration in most years (Bothe and Abraham, 1993). Consequently, water deficits are common

in the WBP, relying on infrequent wet periods every 10-15 years to replenish storage deficits (Marshall et al., 1999; Devito et al., 2005). Historical precipitation data illustrate that rain and snow patterns are variable in the Western Boreal Plain (Table 2; Fig. 3). Total snowfall was near or below average during years during which spring wildfires burnt large areas. Although modest snowfalls are a recurring influence, they do not necessarily dictate fire magnitude; five years with spring wildfires of low burn

area were identified, possessing similar (or lower) total snowfall values than large spring burn area years (Table 2). Earlier snowmelt can extend the dry WBP spring and drying of organic soils, which could therefore extend the period over which spring fires can be generated (Westerling et al., 2006). However, the timing of snowmelt in the WBP does not appear expedited in years of large burned area in the spring, with no significant patterns in the timing of snow-free conditions observed in the seven-year

Gordon Lake snow pillow record (Fig. 3). However, years of high total annual snowfall all align with years with low burned area in the spring (Table 2). This suggests that large SWE can contribute to decreasing the total annual area burned in the spring. Low and infrequent early precipitation events occurred in three of four years with high burned area in the spring. However, due a large proportion of rainfall in continental western Canada generally occurring in summer (Smerdon et al., 2005), dry early

spring is not exceptional, and not restricted to years of high burned area in the spring. The year with the lowest early spring cumulative rainfall in the 20-year record was 2008; however, above average





snowfall and late snow-free conditions decreased wildfire susceptibility in the spring, further

demonstrating the importance of a large snowmelt for reducing wildfire vulnerability (CFRC, 2001).

The 2015-16 hydrological year experienced the second warmest winter temperatures over the

past 20 years. Periodic rises in air temperature above freezing conditions throughout the winter (Fig. 4a)

supplied energy for mid-winter snowmelt and sublimation (Pomeroy et al., 1998), potentially decreasing

available peak SWE for the spring snowmelt period. The modest snowpack melted over a 31-day

period. Immediately following snowmelt, high air temperatures, low relative humidity and high wind

speeds (Fig. 4b; c) created weather conditions optimal for the spread of wildfire (Van Wagner, 1977).

Similar mild winter temperatures and warm, dry spring conditions were present in previous years of

high spring time burned area in 1968, 1998, 2002, and 2011. These years produced fires of a similar

magnitude and total area burned to the Horse River fire of 2016 (Hirsch and de Groot, 1999; Tymstra et

al., 2005; FTCWRC, 2012).

**5.2 Pre-fire hydrology**

A five-year (2011-2016) water table record illustrated the susceptibility of Poplar Fen to extended

drying periods, with years of high spring (2011 & 2016) and summer (2015) burned area corresponding

with low water table position (Fig. 5). At Poplar Fen, water tables also decreased over winter periods in

the absence of precipitation-driven recharge. These prolonged periods of water table decline were

evidenced by logged water table and mineral piezometer observations from the lower and upper fen

areas (Fig. 6). In these areas, hydraulic head in the underlying mineral substratum (~1.5 m b.g.s.)

closely mimicked the pattern of the water table, suggesting that the underlying groundwater at Poplar

Fen is derived mostly from local recharge, rather than from regional groundwater, which would have a





more stable hydraulic head (Siegel and Glaser, 1987). Therefore, peatlands that are supplied mainly by

local groundwater (such as Poplar Fen) become particularly vulnerable to wildfire during high-risk fire

weather conditions (Lukenbach et al., 2017).

   Spring NW fen water table position was also related to the persistence of a frozen upper

5 saturated zone. For example, near surface water tables in fall of 2012 and 2013 (Fig. 5) allowed for

relatively homogenous over winter freezing of the upper saturated zone (Price, 1983), which reduced

the permeability of the peat (Roulet and Woo, 1986; Quinton et al., 2009) and helped store subsurface

water over the winter periods (Price and FitzGibbon, 1987). Ground ice persisted into mid-late May in

2013 and 2014, thus limiting snowmelt water infiltration (Roulet and Woo, 1986) and subsurface water

10 loss to the underlying silt layer (Price and FitzGibbon, 1987). Conversely, the shallow (0-0.2 m) peat

had reached above freezing temperatures by the end of snowmelt (mid-April) in 2016, suggesting that

low (~0.55 m b.g.s.) fall 2015 water tables had prevented the near-surface ground ice. Consequently, the

entire saturated zone was free to recharge the underlying silt layer over the 2015-16 winter, and during

the 2016 snowmelt period, meltwater infiltrated readily to recharge the relatively deep water table.

15 Thus, high antecedent fen water levels provide an important mechanism for overwinter storage and

maintaining higher spring water levels.

   Post-snowmelt 2016, the NW fen water table (0.3 m b.g.s.) was ~0.3 m lower than the water

table observed mid-June, 2011 (Fig. 5), a period without rainfall and with high burned area in the spring

when the 2011 Richardson Fire reached a size similar to the Horse River Wildfire (Pinno et al., 2013).

20 Surprisingly, spring 2016 water tables were more comparable to levels measured in the spring of 2012

(Fig. 5), a year of low burned area in the spring. The lower burned area was likely attributed to larger



and more frequent rainfall events (an additional 14 mm) recorded in the region during the 2012 spring
season. Peatland water table position, therefore, likely cannot serve as a stand-alone metric for
estimating fen wildfire susceptibility in the region, without considering the moisture deficits that can
accumulate above the water table in the absence of precipitation.

Soil moisture in upland duff and margin peat followed a drying trend throughout 2015.
Following snowmelt in 2016, water content in the upland duff and margin peat were not sufficiently
higher than values observed in fall of 2015 (Fig. 7). These data suggest that there was no net wetting to
the organic near surface soils in the upland or margins at Poplar Fen from snowmelt infiltration. This
soil moisture deficit was further enhanced by the lack of spring precipitation and increased evaporative

demand (Hayward and Clymo, 1983) driven by the low humidity, high temperatures and winds at the
time of the Horse River fire (Fig. 4). This deficit would have increased the available fuels for the
wildfire and allowing for significant combustion of these organic layers (Table 4).

**5.3 Assessing the hydrometeorological conditions preceding the Horse River Fire and burning of
Poplar Fen**

The historical meteorological and field hydrological data illustrate the susceptibility of regionally-
abundant WBP peatland watersheds to wildfire during extended dry periods. Results suggest that the
wildfire at Poplar Fen, and the greater Horse River Wildfire, was not simply a consequence of
anomalous drought climate conditions, but rather, interconnected hydrometeorological factors not
uncommon to the Western Boreal Plain, occurring at least twice in the five-year instrument record.

These factors included low autumn soil moisture and water tables, modest snowfall, overwinter
drainage, insufficient spring rainfall, and high spring air temperatures and winds. The synchronicity of



these factors, occurring in the same hydrological year, combined with mature tree stands with high

accumulated fuels ubiquitous to the region, likely contributed to the large magnitude Horse River

Wildfire. The similarities of the hydrometeorological events preceding the Horse River Fire with

previous years (1968, 1998, 2002, and 2011) of similar burned area in the spring (Hirsch and de Groot,

1999; Tymstra et al., 2005; FTCWRC, 2012) suggest that the mild and/or dry fall, winter and spring

conditions conducive for spring fire occur frequently in the region. Moreover, conditions favouring

spring wildfire may be enhanced by climate change, given the responsiveness of forest fuel moisture to

changes in temperature and precipitation (Weber and Flannigan, 1997; Flannigan et al., 2016).

**5.5 Differences in burn severity within Poplar Fen**

During summer 2015, vertical hydraulic gradients decreased in all fen and margin wells over periods of

low precipitation. In lower fen these remained positive throughout the 2015 sampling period (Fig. 6),

indicating upward groundwater discharge into the basal peat (water gain to peatland) from the

underlying silt layer. In upper fen regions, these values were always lower and eventually became

negative over time in the absence of rainfall, suggesting a flow reversal (downward), and loss of water

from the basal peat to the underlying silt layer. Margin areas, located at a higher topographic position

between fen and upland, exhibited the strongest negative hydraulic gradients, suggesting that these areas

were recharging the underlying silt layer throughout the year. These subtle differences in topographic

position therefore played a large role in the observed differences in burn severity between these areas

(Table 4). Hence, treed headwater moderate-rich fens and fen margin tracts in the WBP may be

particularly vulnerable to wildfire.



### 5.6 Soil moisture: an early indicator of spring wildfire danger

The 2015 moisture conditions observed in the upland duff of Poplar Fen were illustrated reasonably

well with the DC. The DC was overwintered for 2016 using a range of startup values from different

methods (Table 3). Scenarios 2 and 3 produced DCs that were lower than the expected DC (scenario 1),

since carry-over and wetting-efficiency coefficients overestimated the recharge to the duff layer by 15-

21%. These default coefficients may not have accounted for the high sublimation rates caused by low

relative humidity and high solar radiation, common to the Western Boreal forests of Canada (Burles and

Boon, 2011). The lower recharge values measured at Poplar Fen (35% of melt water) may also be due

to moisture deficits that accumulated since the summer of 2015, as a high proportion of the available

meltwater went towards recharging the unsaturated mineral soil underlying the duff. The start up DC

that was calculated using the directly measured duff recharge (scenario 4) was much closer to the

expected DC, suggesting that the overwintering calculation is suitable for the duff layer at Poplar Fen

when VWC is directly measured.

      Due to differences in soil bulk density and depth of burn, average duff fuel consumption was

~50% less than the consumption observed in margins (Table 4). The observed duff fuel consumption at

Poplar Fen (~7.0 kg m$^{-2}$), along with the VWC-inferred expected (488) and overwintered (452) final

duff DC values, were both in line with fuel consumption and DC estimates from interior Alaska (Kane

et al., 2007) and are on the higher end of DCs measured from other burned boreal forest fires

throughout continental western Canada (de Groot et al., 2009). Thus, the overwintering procedures that

were calculated using default wetting efficiency coefficients produced lower final DC values that did

not reflect the fuel consumption rates measured at Poplar Fen. The observed range in overwintering DC



calculations in Table 3 highlights the difficulties in determining a proper start up DC for watersheds that experience periods of prolonged drying prior to snowmelt. These overwintering calculations have a substantial impact on DC values calculated for the following growing season, especially in the early spring. Estimations based on VWC measurements may therefore produce more accurate and

conservative spring DC values, given that the selected coefficients may not properly represent the hydrological and meteorological processes occurring in the Western Boreal Plain during the snowmelt period.

**6 Conclusions**

This study applies a combination of pre-fire and historical hydrometeorological data from a moderate-

rich fen watershed, to contextualize the conditions preceding the 2016 Horse River Wildfire. The fire was manifested by dry hydrometeorological conditions extending back to summer 2015. This included low fall soil moisture, modest snowfall, and no spring rainfall, with above-average spring air temperatures and high winds also prevalent; conditions not uncommon in the sub-humid WBP. It was ultimately the less frequent synchronization of these factors that led to a wildfire of this size and

observed depth of burn in boreal forests and wetlands, and the associated fuel losses. These coinciding hydrometeorological conditions share stark similarities with previous years with large burned areas from spring fires, namely 1968, 1998, 2002, and 2011, which may support the notion that fires of this magnitude are a function of WBP climate cycles. However, as natural as these factors may be, spring conditions conducive to wildfire could be enhanced by climate change, given the responsiveness of

these boreal watersheds to changes in temperature and precipitation.



Field data from Poplar Fen confirmed that moisture deficits accumulated between summer 2015 and the Horse River fire the following spring. Following a relatively mild winter, the modest 2016 snowmelt did not raise upland duff and margin peat moisture above fall 2015 values. This was in part due to the hydrogeological setting of Poplar Fen, as water tables and hydraulic head decreased in the

absence of localized precipitation-driven recharge from adjacent uplands, with little or no regional groundwater connection to supplement discharge during extended dry periods. This groundwater connection is further influenced by subtle changes in topographic position, where margins as well as fen areas located at higher elevations are more vulnerable to vertical flow reversals causing groundwater recharge, and experienced greater burn depths and fuel consumption than fen areas at lower topographic

positions. We propose that headwater peatlands in this region fed by localized flow systems will be particularly susceptible to water table fluctuations under a drying climate, rendering them more vulnerable to burning from wildfire.

The dry conditions and subsequent duff fuel consumption observed at Poplar Fen in the spring of 2016 were difficult to illustrate with the Drought Code (DC) when carry-over and wetting-efficiency

coefficients were applied to the overwintering procedure. Closer agreement was found when measured directly measured duff soil moisture recharge was applied to the overwintering procedure in place of the coefficients. These latter DC values were also more comparable to DC values measured in Western Canada and Alaska for similar fuel consumptions, further illustrating the effectiveness of directly measured snowmelt recharge in the overwintering procedure. In order to better gauge the susceptibility

of WBP headwater systems to wildfire in the spring, management strategies could therefore benefit from monitoring soil moisture at different land classes and watersheds. These data would allow for

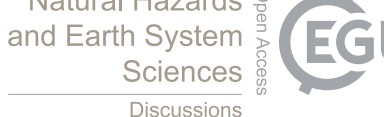

more accurate overwintering DC calculations, and would provide managers more time to prepare for a

fire season by considering additional indicators that can be detected earlier.

## 7 Acknowledgments

The authors wish to thank C. Wells, G. Sutherland, D. Price, E. Kessel, J. Asten, and S. Irvine for their
assistance in the field. We gratefully acknowledge funding from a grant to J.S. Price from the National
Science and Engineering Research Council (NSERC) of Canada Collaborative Research and
Development Program, co-funded by Suncor Energy Inc., Imperial Oil Resources Limited, and Shell
Canada Energy. An additional thank you to Ralph Wright at Alberta Agriculture and Forestry for help
with obtaining historical data, and to Tom Schiks for comments on an earlier version of the manuscript.

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

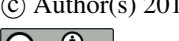



## 9 Figures

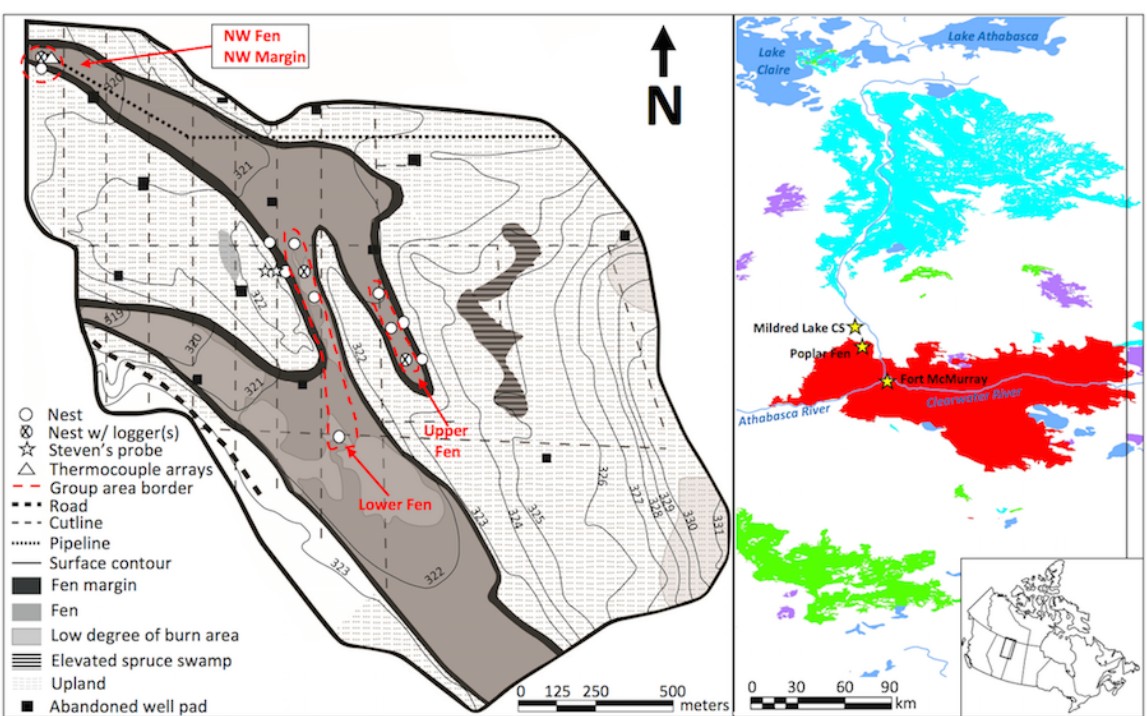

**Figure 1: Map of Poplar Fen watershed (left; 56°56'N; 111°32'W). The entire area was burned with the exception of areas highlighted in blue. Included is an inset of northeastern boreal Alberta (right) showing burned area during years of high spring fire frequency, including 1998 (purple), 2002 (green), 2011 (cyan), and 2016 (red).**

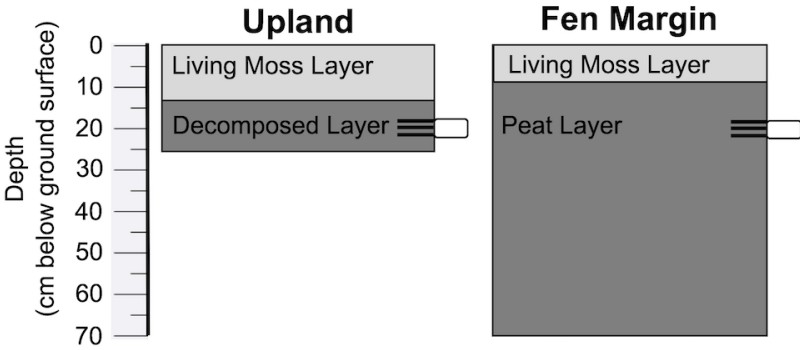

**Figure 2: Moisture probe profiles in upland duff and fen margin peat.**



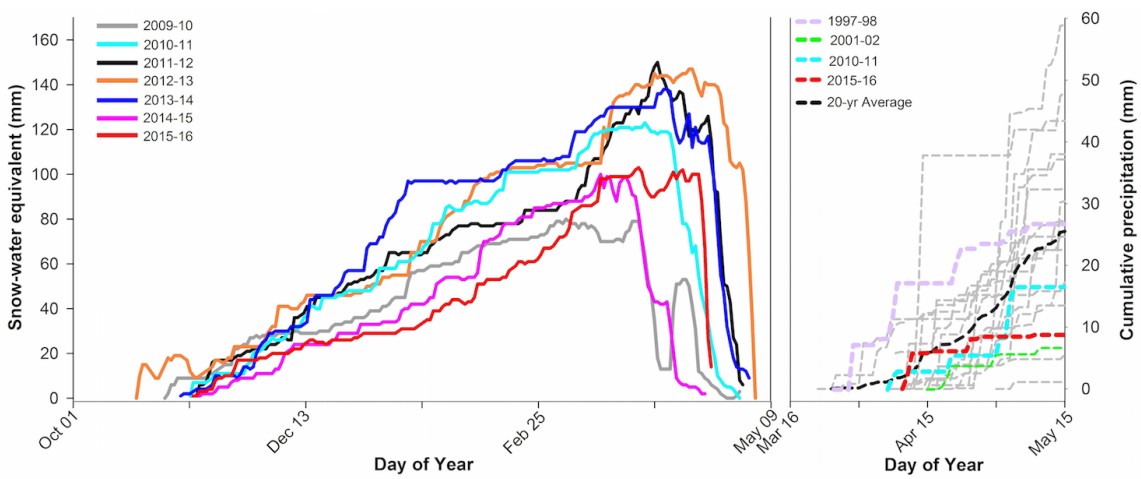

5  **Figure 3: Measured accumulation and ablation of SWE at Gordon Lake snow pillow (left), and interpolated cumulative early spring rainfall from 1996 and 2016 at Poplar Fen (right). Coloured lines in graph b) correspond to years of high burned area in the spring, including 1997-98 (726 968 ha), 2001-02 (496 515 ha), 2010-11 (806 055 ha), and 2015-16 (663 529 ha) (Natural Resources Canada, 2017).**


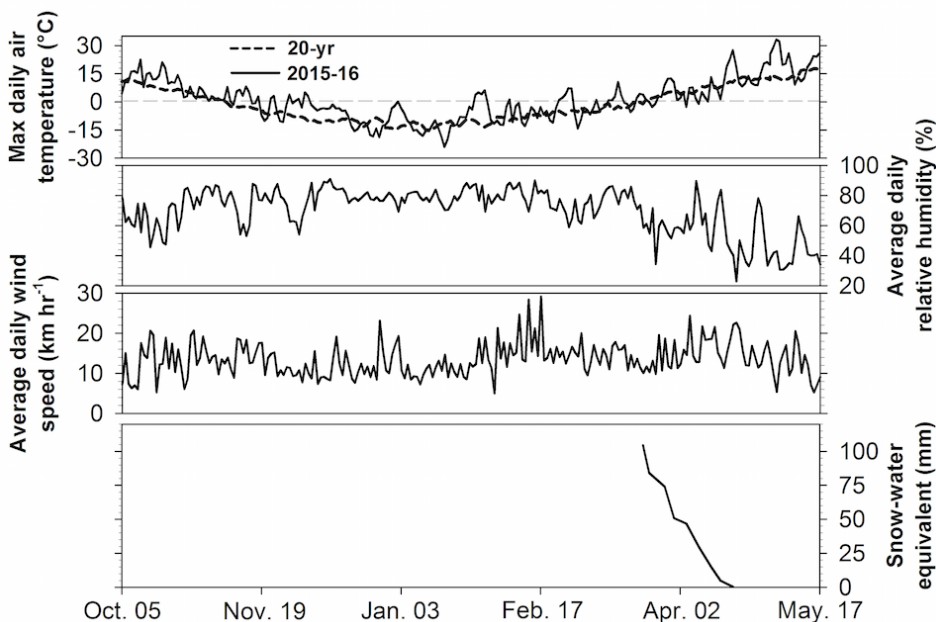

**Figure 4: Daily records of maximum a) air temperature (with 20-yr average), b) average relative humidity and c) wind speed at Mildred Lake climate station from Oct 05 2015 to May 17, 2016, and d) measured area-weighted SWE for Poplar Fen in 2016.**

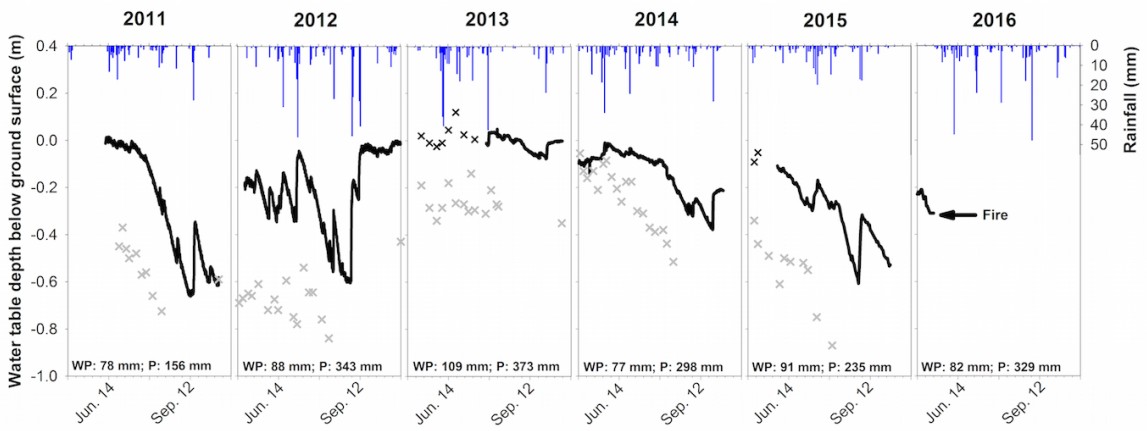

**Figure 5: Logged (lines) and manually (X symbols) recorded water table position at NW fen (black) and NW margin (grey) (see Fig. 1), from 2011-2016, with field-measured rainfall (P), and total winter precipitation (WP) interpolated for the Poplar Fen area.**



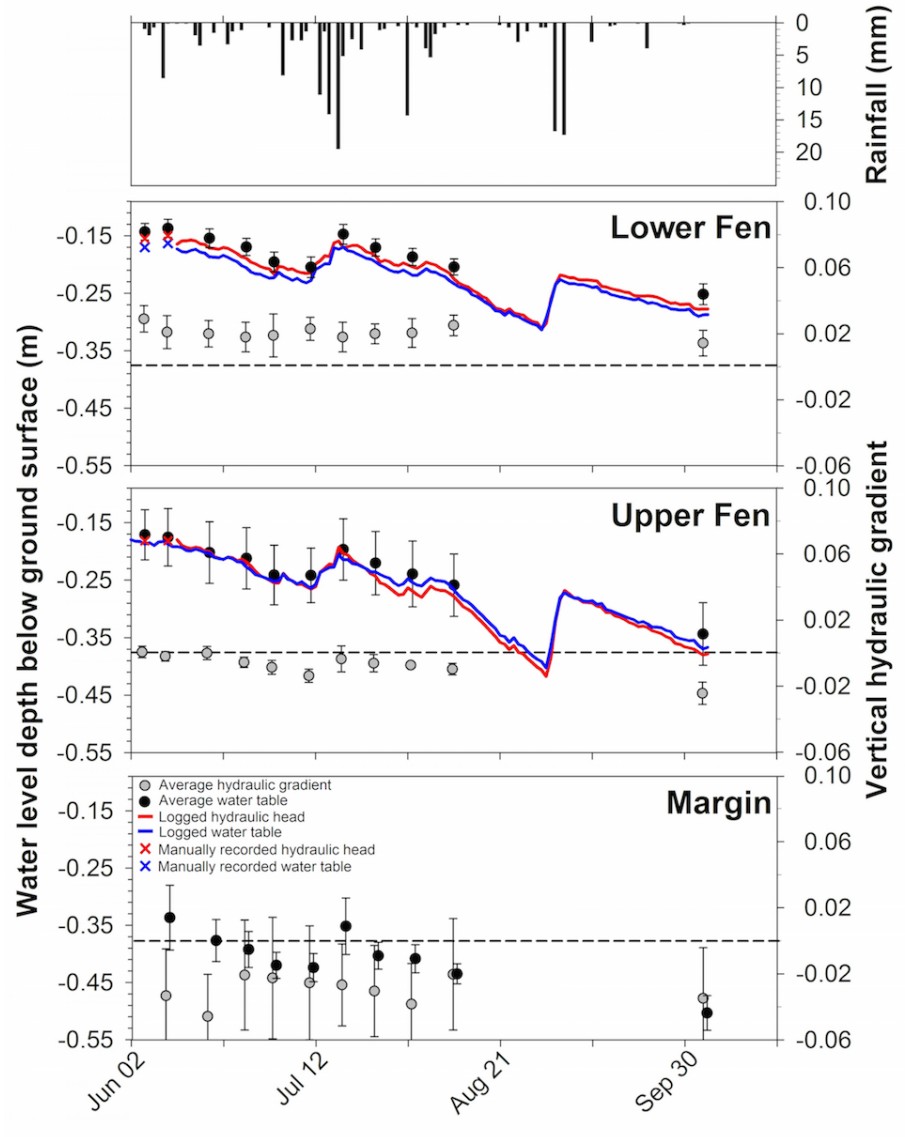

**Figure 6: Average (SE) water table (black x-symbol) and vertical hydraulic gradient (grey circle) between water table and underlying mineral for lower and upper fen, and margin areas, along with logged (line) and manually recorded (x-symbol) water table (blue) and hydraulic head (red) for lower and upper fen areas in 2015. A negative hydraulic represents a loss of water from the fen to the underlying mineral substrates. Rainfall is also illustrated.**




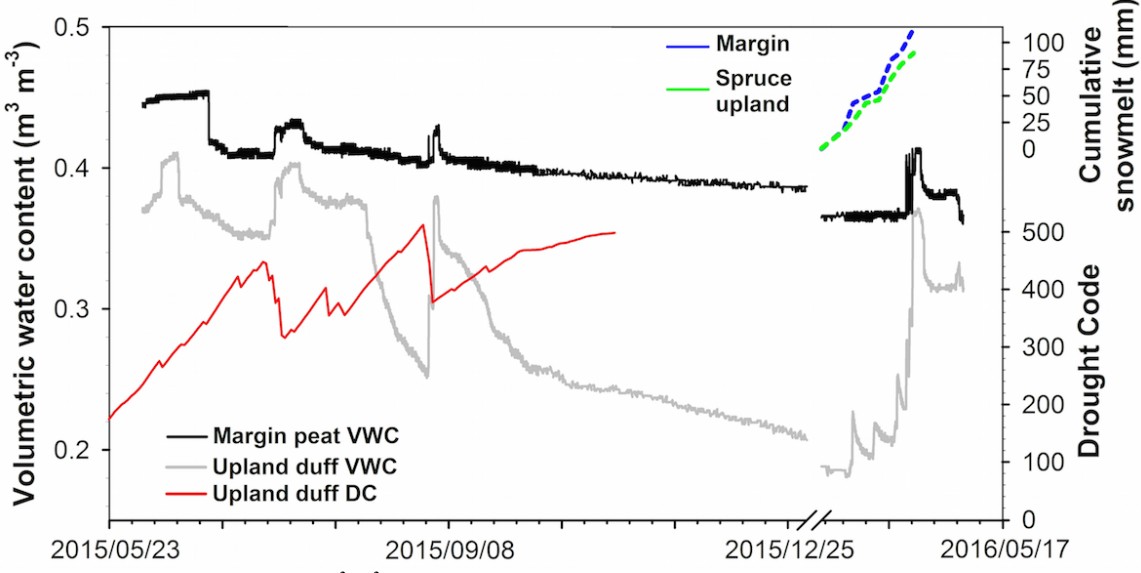

**Figure 7: Volumetric water content (m³ m⁻³) for upland duff and margin peat from Jun 02, 2015-May 02, 2016, including with average 2016 snowmelt recharge (mm) for upland and margin, and Drought Code from May-October, 2015.**

## 10 Tables

**Table 1: Summary of scenarios used for calculating a starting DC for April 19, 2016.**

| Scenario # | Carry-over *a* | Wetting-efficiency *b* |
|---|---|---|
| 1) Expected DC values based on observed relationship between 2015 VWC and DC. | N/A | N/A |
| 2) Overwintering procedure with default CFFDRS values (Lawson and Armitage, 2008) using precipitation from Fort McMurray airport. | 0.75 | 0.75 |
| 3) Overwintering procedure with Alberta Agriculture and Forestry values with Poplar Fen manual SWE data. | 0.5 | 1 |
| 4) Overwintering procedure with upland duff: Using measured 32 mm snowmelt recharge (Oct. 31 – Apr. 19). | 1 | 1 |





**Table 2: Total hydrological year rainfall and snowfall from 1996-2016, interpolated for the Poplar Fen area. Asterisks correspond to years of high burned area in the spring.**

| Year | Total | Rain | Snow |
|---|---|---|---|
| 1996-97 | 467 | 354 | 114 |
| 1997-98* | 265 | 156 | 109 |
| 1998-99 | 280 | 227 | 53 |
| 1999-20 | 395 | 331 | 64 |
| 2000-01 | 356 | 277 | 79 |
| 2001-02* | 396 | 322 | 75 |
| 2002-03 | 424 | 306 | 118 |
| 2003-04 | 396 | 286 | 110 |
| 2004-05 | 523 | 385 | 138 |
| 2005-06 | 409 | 303 | 106 |
| 2006-07 | 352 | 215 | 137 |
| 2007-08 | 387 | 235 | 151 |
| 2008-09 | 269 | 210 | 58 |
| 2009-10 | 421 | 330 | 90 |
| 2010-11* | 235 | 156 | 78 |
| 2011-12 | 430 | 343 | 88 |
| 2012-13 | 481 | 373 | 109 |
| 2013-14 | 375 | 298 | 77 |
| 2014-15 | 326 | 235 | 91 |
| 2015-16* | 412 | 329 | 82 |





5  **Table 3: April 19, 2016 startup and final May 17 DCs for Poplar Fen using four different scenarios.**

| Scenario # | Carry-over a | Wetting-efficiency b | Starting DC on April 19, 2016 | Final DC on May 17, 2016 |
|---|---|---|---|---|
| 1) Expected DC values based on observed relationship between 2015 VWC and DC | N/A | N/A | 357 | 488 |
| 2) Overwintering procedure with default CFFDRS values (Lawson and Armitage, 2008) using precipitation from Fort McMurray airport | 0.75 | 0.75 | 242 | 373 |
| 3) Overwintering procedure with Alberta Agriculture and Forestry values with Poplar Fen manual SWE data | 0.5 | 1 | 212 | 344 |
| 4) Overwintering procedure with upland duff: Using measured 32 mm snowmelt recharge (Oct. 31 – Apr. 19) | 1 | 1 | 321 | 452 |

**Table 4: Average (± SE) surface change and fuel consumption in upland, margin, and fen at Poplar Fen.**

| Type | Measured Ground Surface Change (m) | Pre-Burn Bulk Density (kg m$^{-3}$) | Estimated Fuel Consumption (kg m$^{-2}$) |
|---|---|---|---|
| Duff | 0.10±0.02 | 70 | 7.0±1.2 |
| Margin | 0.13±0.01 | 98 | 13.0±1.2 |
| Fen (burned) | 0.02±0.002 | 70 | 1.6±0.06 |
| Fen (unburned) | 0 | 70 | 0 |

