# Peer review of "Hydrometeorological conditions preceding wildfire, and the subsequent burning of a fen watershed in Fort McMurray, Alberta, Canada"

_Natural Hazards and Earth System Sciences, 2017_

## Referee Comment (RC1) · Anonymous Referee #1 · 28 Oct 2017

Review of nHESS-2017-312

The manuscript "Hydrometeorological conditions preceding wildfire, and the subsequent burning of a fen watershed in Fort McMurray, Alberta, Canada" provides a very thorough investigation of the hydrometeorological conditions and some of the burn parameters (depth of burn and fuel consumption) of a high-profile wildfire that led to the greatest financial wildfire losses in Canadian history. It is therefore of substantial interest well beyond the general readership of the journal.

The study has been conducted very thoroughly, the interpretations have been made very carefully and the manuscript is very well written. I have therefore only have a

few minor suggestions and requests for clarification that I feel should be addressed to further strengthen the manuscript.

Introduction/Discussion: It would be useful for the international reader, particularly when considering those who may read it in future years, to set the Horse River fire into the wider context of burning in this wider region. I.e. some lines on ignition sources, area burned, burn depth and duff fuel consumption observed in other fires and years in this region. You could also state the evacuation need and financial losses that made this fire so high profile (e.g. http://www.ibc.ca/bc/resources/media-centre/media-releases/northern-alberta-wildfire-costliest-insured-natural-disaster-in-canadian-history).

Methods: a) Burn depth was assessed based on survey data from well stick-up (length of PVC above ground surface). This is a little unclear. Would PVC length not have been potentially affected by burning? Is DOB determined at monitoring sites not affected by the fact that organic soils may have been somewhat compressed/disturbed compared here. Have there not been other systematic ground surveys of DOB?

b) Hydrometeorological data averages have been derived from 1996-2016 and are compared to 2015/16 data. Would it not be more meaningful to compare the 2015/16 situation with the average of the preceding period rather than including it when calculating the average?

Discussion/Conclusion: These sections have been phrased very carefully and are fully supported by the data. To strengthen the implications of the work, however, it would be very useful to provide a quantitative estimate of how frequent the synchronisation of these hydrometeorological factors may be in this region. What is their likely return interval under current and perhaps even future climatic conditions in this region?

---

## Referee Comment (RC2) · Anonymous Referee #2 · 31 Oct 2017

[referee-annotated manuscript omitted]

---

## Author Comment (AC1) · 8 Nov 2017

Thank you for taking the time to review our manuscript. We found the comments very helpful and have no serious issues with them. Please find our responses to your attached comments below.

Regards,

- Matthew Elmes, et al.

Introduction/Discussion: It would be useful for the international reader, particularly when considering those who may read it in future years, to set the Horse River fire

into the wider context of burning in this wider region. I.e. some lines on ignition sources, area burned, burn depth and duff fuel consumption observed in other fires and years in this region. You could also state the evacuation need and financial losses that made this fire so high profile (e.g. http://www.ibc.ca/bc/resources/media-centre/media- releases/northern-albertawildfire- costliest- insured-natural- disaster-in-canadian-history).

- These are very good points, and we agree that some more context would be appropriate for those who may not be fully aware of the significance of the fire. We will add more context into our next version.

Methods: a) Burn depth was assessed based on survey data from well stick-up (length of PVC above ground surface). This is a little unclear. Would PVC length not have been potentially affected by burning? Is DOB determined at monitoring sites not affected by the fact that organic soils may have been somewhat compressed/disturbed compared here. Have there not been other systematic ground surveys of DOB?

- To be more specific, we compared our pre-burn ground surface (top of pipe elevation – depth to ground) to our post-burn ground surface (top of pipe elevation – depth to ground). It is important to note that we completed DGPS surveys in the fall of both 2015 and 2016, so even though the pipes burned, we were able to determine burn depths by comparing surveys. We will make sure we describe this in better detail in the next MS version.

b) Hydrometeorological data averages have been derived from 1996-2016 and are compared to 2015/16 data. Would it not be more meaningful to compare the 2015/16 situation with the average of the preceding period rather than including it when calculating the average?

- Yes this would make more sense. We found it important to identify where the four most recent high-burn years fell within the 20-yr historical record. However, when comparing the 2015-16 winter temperatures to the average, it would make more sense to compare

it to 1996-2015. We will change this for the next version.

Discussion/Conclusion: These sections have been phrased very carefully and are fully supported by the data. To strengthen the implications of the work, however, it would be very useful to provide a quantitative estimate of how frequent the synchronisation of these hydrometeorological factors may be in this region. What is their likely return interval under current and perhaps even future climatic conditions in this region?

- Considering this synchronization has happened four of the past 20 years. It would suggest that our current recurrence interval would be every five years. However, it would be difficult to quantify how this will change in the future given the uncertainties regarding climate change. We can only speculate that it will become more frequent, and any actual predictions may be outside the scope of the paper.

---

## Author Response (AR1)

To Mario Parise, Editor, NHESS

On behalf of me and my co-authors, thank you for your prompt response following the peer-review of our article titled "Hydrometeorological conditions preceding wildfire, and the subsequent burning of a fen watershed in Fort McMurray, Alberta, Canada". We have carefully reviewed all comments from both referees and have incorporated edits into our manuscript. Please see attached responses. We incorporated most of the changes, however, rebutted a few of them. Please let us know if there are any other recommendations necessary for publication. We have also attached two versions of the manuscript, one with the changes highlighted (see supplement) and one without. The images attached at the bottom of the manuscript are decreased in resolution. Please see the attached compressed folder with the high-resolution figures. These files have also been updated so some of the photos will be different than the original compressed image file from the original submission.

Regarding your comment related to debris flow, it is indeed a question worth pursuing. In fact, I am currently working on a study to identify changes to the hydrophysical properties of upland soils in the AOSR following wildfire. Specifically, I wish to better understand how fire effects upland soil hydrophobicity, water retention, and storage, and what these implications are for the hydrologic regime and water balance of the Poplar Fen watershed. We speculate that wildfire will not significantly enhance debris flow at watersheds similar to Poplar Fen, and this is primarily due to the sub-humid climate as well as the low relief topography in the region, as runoff is not a common occurrence on these gently sloping coarse-grained uplands that tend to have a large storage potential. That being said, the floodplain regions of the Athabasca River have much steeper slopes, and many of these areas were impacted by the fire. We would expect significant increases in debris flow given the destruction caused to the vegetation and soils, however, we are not fully aware of the research done on this in the region. I appreciate you acknowledging the importance of this information and I hope to let you know of my findings in the near future.

If you require any other information, please feel free to contact me directly.

Kind Regards,

Matthew Elmes, M.Sc. Ph.D. Candidate
Department of Geography and Environmental Management
University of Waterloo
Waterloo, ON, N2L3G1
Canada

**Anonymous Referee #1**

Review of NHESS-2017-312

> "The manuscript "Hydrometeorological conditions preceding wildfire, and the subsequent burning of a fen watershed in Fort McMurray, Alberta, Canada" provides a very thorough investigation of the hydrometeorological conditions and some of the burn pa- rameters (depth of burn and fuel consumption) of a high-profile wildfire that led to the greatest financial wildfire losses in Canadian history. It is therefore of substantial interest well beyond the general readership of the journal.
>
> The study has been conducted very thoroughly, the interpretations have been made very carefully and the manuscript is very well written. I have therefore only have a few minor suggestions and requests for clarification that I feel should be addressed to further strengthen the manuscript."

**Thank you for taking the time to review our article. You will find our responses to your comments below in bold. Please let us know if you are not fully pleased with our changes and we are open to revisiting them.**

**Regards,**

- **Matthew Elmes et al.**

Introduction/Discussion: It would be useful for the international reader, particularly when considering those who may read it in future years, to set the Horse River fire into the wider context of burning in this wider region. I.e. some lines on ignition sources, area burned, burn depth and duff fuel consumption observed in other fires and years in this region. You could also state the evacuation need and financial losses that made this fire so high profile (e.g. http://www.ibc.ca/bc/resources/media-centre/media-releases/northern-alberta- wildfire-costliest-insured-natural-disaster-in-canadian-history).

**- Ignition sources are already provided on the third paragraph of the introduction.**

**- We have included a 10-yr average area burned for Alberta for context.**

**- Further context added for burn depth and duff fuel consumption observed in other fires and years in this region, as well as the destructive nature of fire and need for evacuation.**

Methods: a) Burn depth was assessed based on survey data from well stick-up (length of PVC above ground surface). This is a little unclear. Would PVC length not have been potentially affected by burning? Is DOB determined at monitoring sites not affected by the fact that organic soils may have been somewhat compressed/disturbed compared here. Have there not been other systematic ground surveys of DOB?

**Thank you for bringing this up. We have changed the wording here to inform the reader**

**that DGPS surveys were conducted pre- (2015) and post-fire (2016). We inferred a pre- and post-surface elevation by subtracting the distance to ground from the top of pipe elevation, then compared these values between years.**

b) Hydrometeorological data averages have been derived from 1996-2016 and are compared to 2015/16 data. Would it not be more meaningful to compare the 2015/16 situation with the average of the preceding period rather than including it when calculating the average?

**Not only are the 2015-16 data compared to the 20-yr average, but so are the other three years of high spring burned area. For this reason, we believe it is important to keep 2015-16 along with the other three years in the 20-yr average. We could separate the data into two groups with two separate averages, however, this may not be as scientifically sound as we would like to cover a wide range of hydrometeorological conditions to create meaningful averages.**

**That being said, we agree with comparing the 2015-16 winter air temperatures with the previous 19 years and have changed that in our results section.**

Discussion/Conclusion: These sections have been phrased very carefully and are fully supported by the data. To strengthen the implications of the work, however, it would be very useful to provide a quantitative estimate of how frequent the synchronisation of these hydrometeorological factors may be in this region. What is their likely return interval under current and perhaps even future climatic conditions in this region?

**For the first suggestion, our analyses suggest a return interval of five years, and we have changed out MS to explicitly state this. Regarding predictions for the future, this may be outside of the scope of our paper. It may also be unnecessary as Mike Flannigan out of University of Alberta has spent considerable time and effort on understanding changes in wildfire frequency, magnitude, and fire season length, due to climate change. Our results do not contradict his findings, and at this point we can only speculate that the frequency of synchronization will increase under a warmer and drier future climate.**

**Anonymous Referee #2**

I liked the paper. Is very well written and easy to follow. The results were quite expectable, but it is a good contribution. Please see my comments in a PDF attached

Please also note the supplement to this comment: https://www.nat-hazards-earth-syst-sci-discuss.net/nhess-2017-312/nhess-2017-312- RC2-supplement.pdf

**Thank you for taking the time to review our manuscript. Since your comments were included in a PDF, we have responded to your comments by pasting them into this document adding additional text to each of your comments (refer to my initials for my response – MCE). Please let us know if you are not fully pleased with our changes and we are open to revisiting them.**

**Regards,**

**Matthew Elmes et al.**

Abstract line 6 suggestion: accepted and changed

Abstract line 10: "by fire?"
- MCE - Yes will change the writing

Pg. 2 line 25 wording suggestion: accepted and changed

Pg. 2 line 26 comment: "in water?"
- MCE - This is explained in the next sentence, and **we have also added species richness to make it less confusing**. But no it is not entirely determined by water chemistry.

Pg. 3 line 3. "Does human behavior have something to do with this as well?"
- MCE - Yes, obviously. However, human behaviour will cause ignition sources regardless of the time of year. The susceptibility will therefore be determined by the hydrometeorological conditions.

Pg. 3 line 12. "This reference is quite old. Is it possible to provide data from recent years."
- MCE - Yes thank you for pointing this out. We will also add to following citation:
**Podur J and Wotton B M 2010 Will climate change overwhelm fire management capacity? Ecol. Model. 221 1301–9.**

Pg. 5 line 5: "Study the hydrological and meteorological conditions pre-fire is very important. This has to be placed in a wider context to be better justified."
- MCE - Here we are not talking about our study, but rather, we are commenting on how the fire motivated scientists in the area to better understand these conditions. This is setting up the need for our study, which we introduce in the next paragraph. **Please re-consider this comment.**

Pg. 5 line 16: "Please move this after the line 8."

- MCE – Changed

Pg. 6 line 12: "Please show the soil types of the studied areas."

- MCE - Are you asking for a quantification of grain size? Or do you specifically want us to mention the soil types in the upland? If it is the latter, uplands are primarily composed of Brunisols, with a band of Luvisol soil in the riparian areas between upland and margin. We have added the latter to the site description. See highlighted text in updated MS.

Pg. 7 line 7: "Interpolate data with 2-7 samples is to little to have a good prediction."

- MCE - Although this true, the greatest importance in the IDW was given to a station located >10 km from Poplar Fen. We saw close agreement between the raw data from this climate station, and the data interpolated for our region. We also saw good agreement between daily precipitation values in the IDW and our tipping bucket data at Poplar Fen for the same days.

Pg. 16 line 3: "You should calculate the coefficient of correlation, not the coefficient of determination. Show the p value as well."

- MCE - changed and updated.

Pg. 23 line 8: "The conclusions should be reduced and highlight only the most important results."

- MCE - We shortened all three paragraphs.

[revised manuscript text omitted]